# Multiple Salivary Proteins from *Aedes aegypti* Mosquito Bind to the Zika Virus Envelope Protein

**DOI:** 10.3390/v14020221

**Published:** 2022-01-24

**Authors:** Paola Carolina Valenzuela-Leon, Gaurav Shrivastava, Ines Martin-Martin, Jenny C. Cardenas, Berlin Londono-Renteria, Eric Calvo

**Affiliations:** 1Laboratory of Malaria and Vector Research, National Institute of Allergy and Infectious Diseases, National Institutes of Health, Rockville, MD 20852, USA; paolacarolina.valenzuelaleon@nih.gov (P.C.V.-L.); gaurav.shrivastava@nih.gov (G.S.); martinines@hotmail.com (I.M.-M.); 2Arbovirology Laboratory, Department of Tropical Medicine, Tulane University, New Orleans, LA 70112, USA; carocardenasg@hotmail.com (J.C.C.); blondono@tulane.edu (B.L.-R.)

**Keywords:** arboviruses, blood feeding, saliva, arthropod, salivary glands, vector borne diseases, antibody-dependent enhancement

## Abstract

*Aedes aegypti* mosquitoes are important vectors of several debilitating and deadly arthropod-borne (arbo) viruses, including Yellow Fever virus, Dengue virus, West Nile virus and Zika virus (ZIKV). Arbovirus transmission occurs when an infected mosquito probes the host’s skin in search of a blood meal. Salivary proteins from mosquitoes help to acquire blood and have also been shown to enhance pathogen transmission in vivo and in vitro. Here, we evaluated the interaction of mosquito salivary proteins with ZIKV by surface plasmon resonance and enzyme-linked immunosorbent assay. We found that three salivary proteins AAEL000793, AAEL007420, and AAEL006347 bind to the envelope protein of ZIKV with nanomolar affinities. Similar results were obtained using virus-like particles in binding assays. These interactions have no effect on viral replication in cultured endothelial cells and keratinocytes. Additionally, we found detectable antibody levels in ZIKV and DENV serum samples against the recombinant proteins that interact with ZIKV. These results highlight complex interactions between viruses, salivary proteins and antibodies that could be present during viral transmissions.

## 1. Introduction

Mosquitoes are vectors of arboviruses considered as a global health problem. In early 2015, the Zika virus (ZIKV) outbreak occurred and rapidly became a worldwide concern due to the severe complication of microcephaly and other fetal malformations in Brazil [1]. Zika virus belongs to the Flaviviridae family along with Yellow Fever virus (YFV), Dengue virus (DENV), West Nile virus (WNV), Japanese Encephalitis virus (JEV), and Tick-borne encephalitis virus (TBEV), a group of viruses that induces a range of severe disease symptoms including hemorrhagic fever or encephalitis in humans [2,3]. These viruses are globally distributed and poses serious global public concern with few successful vaccines available worldwide. Analyzing the transmission mechanisms of these viruses could be important to understand the spread of these viruses; however, studies are still elusive.

The transmission of these viruses is initiated when an infected female mosquito bites and releases virus along with saliva into the skin of the human host. The saliva of mosquitoes is rich in anti-clotting, anti-platelet, and vasodilatory compounds that help them during the blood-feeding, suppressing the hemostasis, inflammation, and immune response of the host [4]. In addition to these direct effects, mosquito saliva can also indirectly affect the transmission of pathogens. Previous studies have demonstrated that mosquito saliva can facilitate viral transmission and contribute to the pathogenesis of the disease. The saliva of the *Aedes aegypti* mosquito increases the viremia and severity in DENV, WNV, Rift Valley virus (RFV), and Semliki Forest virus (SFV) infection in mice [5,6,7,8]. Similar observations have been found with DENV, WNV, and Chikungunya virus (CHIKV) infections using skin cells such as keratinocytes and fibroblasts. In these studies, mosquito saliva decreases the expression of type I interferon genes (IFNs) and interferon-stimulated genes (ISGs) that have antiviral activity [9,10,11].

Salivary components of *Aedes aegypti* mosquitoes have been identified as significant contributors to increased viral replication. A salivary protein named “34 kDa protein” increases DENV viral titer in human keratinocytes, reducing the expression of the antimicrobial peptides LL-37 and S100A7 and type I interferons [12]. In ZIKV infection, a salivary protein named LTRIN facilitates the transmission of the virus and increases the severity of the disease, interfering with signaling via the lymphotoxin-β receptor in vitro and in vivo models [13]. Recently, the venom allergen-1 (Aava-1) from *Aedes aegypti* was found to promote flavivirus transmission through the activation of host autophagy in host immune cells of the monocyte lineage [14]. In addition, several *Aedes aegypti* salivary proteins have been reported to directly interact with virions, using virus overlay protein binding assays (VOPBA) [15,16] and tandem mass spectrometry; two proteins that interact with DV have been identified as a one D7 long-form and an apyrase [16,17]. All these studies have pointed out the role of salivary proteins in promoting flavivirus transmission and pathogenesis in bitten hosts. Nonetheless, more mechanisms of salivary proteins in flavivirus transmission remain to be understood.

In this study, we cloned and expressed 21 recombinant proteins from *Aedes aegypti* salivary glands. Using surface plasmon resonance, we identified salivary proteins that interact with the envelope protein of the ZIKV (ZIKV-E). Although these interactions do not affect viral replication in cultured endothelial cells and keratinocytes, we found that proteins interacting with ZIKV are immunogenic in ZIKV and DENV serum samples. These results suggest that mosquitoes release viral particles coated with salivary gland proteins into the host’s skin. We also found that antibodies from patients infected with DENV and ZIKV can recognize the virus–salivary protein complex. This work offers valuable insight into the possible interactions of the virus with mosquito salivary proteins.

## 2. Results

### 2.1. Expression and Purification of Recombinant Salivary Proteins from Aedes aegypti Mosquitoes

To investigate whether mosquito salivary proteins interact with ZIKV and have a role in viral infection, we selected several mosquito salivary proteins to be evaluated in our assays; like many blood-feeding arthropods, male mosquitoes do not feed on blood and may not be directly involved in pathogen transmission. We focused our efforts on salivary proteins expressed in female mosquitoes. This sex and tissue specificity allows us to narrow down candidate proteins with possible pathogen transmission functions. The selection criteria for protein expressions included presence of secretion sequence and expression in the salivary gland of female mosquitoes only. Based on these criteria, we selected twenty-one proteins (Appendix A) that were cloned in VR2001, expressed in HEK293E cells and purified by affinity and size exclusion chromatography. After the final purification step, the proteins were visualized as a single band by Coomassie-staining gel electrophoresis (Figure 1). The identities of purified recombinant proteins were confirmed by N-terminal sequencing.

### 2.2. Recombinant Salivary Proteins from Aedes aegypti Mosquitoes Bind to ZIKV-E and ZIKV-VLPs

Numerous studies have demonstrated that mosquito saliva can facilitate viral transmission and contribute to the subsequent disease increase [6,8,10,11,14,18]. Nonetheless, the underlying mechanisms of salivary proteins in flavivirus transmission are not fully understood. Surface plasmon resonance (SPR) assays were designed to investigate whether or not the recombinant proteins from *Aedes aegypti* bind to ZIKV-E protein. For SPR analysis, ZIKV-E protein was immobilized on a CM5 sensor chip at 1500 response unit (RU), and 200 nanomolar (nM) of the recombinant mosquito salivary proteins were flowed over the sensor surface (Figure 2A). Six recombinant salivary proteins (AAEL000793, AAEL008766, AAEL007420, AAEL006347, AAEL024303, and AAEL006417) showed a positive binding with ZIKV-E protein (Figure 2B). Next, we calculated the affinity constants for the binding of ZIKV-E protein with the recombinant AAEL000793, AAEL007420, and AAEL006347 by kinetics assays. These three proteins showed a higher response unit in the first screening analysis. In this experiment, the ZIKV-E protein was used as a ligand, and the recombinant AAEL000793, AAEL007420, and AAEL006347 were flowed over the sensor for 150 s at concentrations ranging from 0.015 nM to 500 nM (serial dilutions). We were able to determine the dissociation constants (K_D_) and the resulting sensograms were fitted using a 1:1 binding model (Figure 2C). The calculated K_D_ value for each protein is indicated in Table 1. These data show that AAEL000793, AAEL007420, and AAEL006347 directly interact with the ZIKV-E protein.

To confirm this finding, we also analyzed the binding by ELISA assays. We immobilized the Zika virus-like particles (ZIKV-VLPs) on an ELISA plate and incubated them with the recombinant mosquito salivary proteins. To detect the binding, we use an antibody against the polyhistidine tag (His-Tag-Ab) present in all the recombinant proteins but not in the ZIKV-VLPs. Our results show an interaction between the ZIKV-VLPs and the recombinant AAEL007420 and AAEL006347 but not with AAEL000793 and AAEL003601 (Figure 2D). Negative interaction using the AAEL003601 correlates with the results obtained in the SPR screening where no binding was detected (Figure 1). The binding of AAEL000793 was tested again by ELISA using a mouse polyclonal antibody against the recombinant protein (AAEL000793-pAbs). For these experiments, AAEL000793 was incubated over ELISA plates previously coated with ZIKV-VLPs or ZIKV-E protein. ELISA binding assay using the AAEL000793-pAbs antibodies showed a positive binding of the AAEL000793 with ZIKV-VLPs and ZIKV-E protein (Figure 2E). The finding in this study indicates that AAEL000793, AAEL007420, and AAEL006347 interact with the ZIKV-E protein and ZIKV-VLPs.

### 2.3. Proteins That Interact with the ZIKV-E and ZIKV-VLPs Do Not Affect the Viral Replication in In Vitro Cell Cultures of Endothelial Cells and Keratinocytes

To test if the proteins that interact with the ZIKV-E and ZIKV-VLPs affect viral infection, Human Primary Dermal Microvascular Endothelial Cells (HDMVECn) and Primary Epidermal Keratinocytes (HEKn) were infected with ZIKV at 5 multiplicities of infections (MOIs) for 24 h along with 200 nM of recombinant AAEL000793, AAEL007420, and AAEL006347. After 24 h post infection, plaque assays were performed to detect viral replication (Figure 3). Our results show that the recombinant proteins do not affect the in vitro viral replication in endothelial cells and keratinocytes when compared to cells infected with ZIKV alone.

### 2.4. Production of Polyclonal Antibodies againtst Recombinant Mosquito Salivary Proteins

Mosquito salivary proteins are immunogenic, and antibodies against them have been suggested as potential vaccine candidates [19,20,21,22]. We found that salivary proteins interacting with the ZIKV-E protein and ZIKV-VLPs are immunogenic in mice. BALB/c mice were inoculated with the recombinant AAEL000793, AAEL006347 or AAEL003601; the serum samples were tested by ELISA for the presence of antibodies against the recombinant salivary proteins. All recombinant salivary proteins tested here produced detectable antibody response in mice (Figure 4).

### 2.5. Mosquito Salivary Proteins That Interacting with ZIKV-E and ZIKV-VLPs Virus Are Immunogenic in ZIKV and DENV Serum Samples

The three recombinant salivary proteins that interact with the ZIKV produced an antibody response in mice. We next sought to confirm whether the recombinant salivary proteins are immunogenic in patients with ZIKV and DENV. We measured the levels of anti-IgG antibodies against the recombinants AAEL000793, AAEL007420, and AAEL006347 in serum samples of patients infected with ZIKV or DENV, using nonreactive to the salivary proteins as a negative control. Samples from DENV and ZIKV patients were collected during the acute and early convalescence phase of DENV and ZIKV. We found detectable antibody levels in ZIKV serum samples against the recombinants AAEL007420 and AAEL006347 compared to our control samples (Figure 5A). DENV serum samples showed significant antibodies levels against the recombinants AAEL000793, AAEL007420, and AAEL006347 (Figure 5B). Our results suggest that complexes of the virus, salivary proteins and antibodies against the virus and salivary proteins could be present during natural virus transmission.

## 3. Discussion

During the ZIKV transmission, the mosquito releases the virus in the saliva along with the salivary proteins. These mosquito salivary proteins impact cellular and immunological responses in the host, affecting viral replication [5,6,8,9,23,24,25]. Some components in mosquito saliva have been described as facilitators or inhibitors of viral replication in vitro and in vivo experiments [12,14,17,18]. However, the mechanism behind these effects remains unclear. Our findings demonstrate novel interactions between mosquito salivary proteins and ZIKV; these interactions do not affect the viral replication in HDMVECn and HEKn cells; however, salivary proteins that interact with the ZIKV-E protein were recognized by IgG antibodies in serum samples from patients with DENV and ZIKV infection.

In the current study, we evaluated whether proteins in the salivary gland of mosquitoes *Aedes aegypti* interact with the ZIKV-E protein and whether this interaction affects viral replication. In order to identify the salivary proteins that interact with the virus, twenty-one proteins were cloned into an expression vector and expressed in mammalian HEK293E cells; these proteins are female-specific, secreted in the salivary glands, and have a secretion sequence. Initially, in our SPR results, we detected six recombinant salivary proteins that interacted with the ZIKV-E protein. Moreover, we performed a kinetic binding assay for the three proteins that showed the highest response in the screening analysis by SPR. These proteins corresponded to the Antigen 5 protein (AAEL000793), Factor Xa inhibitor (AAEL007420), and Apyrase (AAEL006347). We determined nanomolar affinity binding levels between the recombinant salivary proteins AAEL000793, AAEL007420, and AAEL006347 with the ZIKV-E protein, as we showed in Table 1. No known function has been described for AAEL000793 [26]; protein AAEL007420 is a serin protease, homolog of Alboserpin protein present in *Aedes albopictus*. Alboserpin has been reported as a tight inhibitor of Factor Xa with antithrombotic activity [27]. AAEL006347 is an apyrase that has high expression levels in the salivary gland of *Aedes aegypti* mosquitoes; this protein hydrolyzes ATP and ADP, inhibiting ADP-dependent platelet aggregation [28,29]. The binding of this apyrase with DENV virus has been reported before by Virus Overlay Protein Binding Assays (VOPBA) [16]. Here, we used a highly sensitive surface plasmon resonance technique to clearly demonstrate the interaction of these three salivary proteins with the envelope protein of ZIKV. In addition, we provided evidence that these proteins interact with the viral particles by ELISA binding assays. In the case of the AAEL000793, the binding was only detected using a polyclonal antibody against the recombinant protein. No binding was detected using His-Tag-Ab. Interestingly, AAEL000793 was recently found to enhance DENV and ZIKV infections through activation of autophagy mechanisms in THP-1 cells (acute monocytic leukemia cell line) [14]. These authors also found that antibodies against V5- Tag were unable to detect it in ELISA. Taken together, these results indicate that affinity tags at the carboxy-terminal end of the protein is not exposed under native condition and cannot be detected by anti-6xHis or V5 antibodies.

We also tested the effect of these three proteins in viral replication by measuring the levels of infective viral particles present in the supernatants of HDMVECn and HEKn cells by plaque assays after 24 h post-infection. No significant differences in viral particles were observed in cell supernatants from cells infected with either ZIKV alone or with 200 nM of AAEL000793, AAEL007420, or AAEL006347. We concluded that the interaction of the mosquito salivary proteins with ZIKV does not necessarily promote or inhibit viral titers in these primary cells, and recent studies have shown that the effect of salivary proteins in arbovirus infection is relevant in vivo, but not in vitro or ex vivo [30]. In the case of DENV, one D7 protein interacts with the viral particles and decreases the viral infection in in vivo and ex vivo experiments [17]. It would be interesting to assess the effects of the salivary proteins in another cellular linage and animal models. Moreover, we analyzed if these proteins were immunogenic in serum samples from DENV and ZIKV patients. Our results demonstrate that DENV and ZIKV patients have antibodies against the proteins AAEL000793, AAEL007420, and AAEL006347 when we compared with the negative controls (human serum samples without antibodies against the salivary proteins).

Results from previous studies have shown that mosquito salivary proteins are immunogenic, and some of the components have been proposed as immunoepidemiologic markers to evaluate human exposure to mosquito bites [31,32,33], yet little is known about the molecular mechanism of how anti-salivary protein antibodies affect viral transmission [34,35,36]. While it has been reported that serum samples from patients diagnosed with severe DENV disease have more antibodies against the apyrase protein compared to patients without severe DENV fever or infection [37], more research is needed to elucidate the role of antibodies against salivary proteins in viral transmission.

Recently antibody-dependent enhancement (ADE) phenomenon has been reported in skin explants with DENV and ZIKV infections. In ADE, non-neutralizing antibodies enhance the virus entry to the cell and replication. Pre-existing heterotypic antibodies can enhance DENV and ZIKV infection and replication in human skin [38]. Furthermore, people who live in endemic areas of DENV and ZIKV with active transmission have antibodies against mosquito salivary proteins [35,36]. Pre-existing antibody responses against mosquito salivary proteins have been proposed as a potential vaccine for vector-borne diseases [19,21,39]. However, in some cases, the presence of antibodies against SGE can favor the virus transmission modulating the host immune response [40,41]. In DENV infection, the pre-exposure to *Aedes aegypti* SGE enhanced the infection over ADE conditions [42]. We hypothesize that during the ZIKV transmission, virus–salivary protein complexes and antibodies against virus and/or salivary proteins may impact viral infection and transmission in the field.

In conclusion, our findings demonstrate that salivary proteins can directly interact with viral envelope proteins as well as intact viral particles. The interaction of the salivary proteins with the virus does not necessarily impact viral replication. Mosquito salivary proteins interacting with the virus (AAEL000793, AAEL007420, and AAEL006347) are immunogenic in DENV and ZIKV patients used in this work. Based on our results, we hypothesized that infected mosquito could release viral particles coated with mosquito salivary proteins during natural transmissions. Antibodies against these salivary proteins might mediate ADE entry of ZIKV. Future studies that take these variables into account have the potential to elucidate the role of mosquito salivary proteins and antibodies in viral transmission.

## 4. Materials and Methods

### 4.1. Protein Purification

All of the DNA sequences were codon-optimized for mammalian expression systems and synthesized by BioBasic Inc. VR2001-TOPO (Vical Incorporated, San Diego, CA, USA). The vector harboring the protein sequences were transformed in One Shot TOP10 Chemically Competent *E. coli* (Invitrogen, Waltham, MA, USA). Recombinant protein expression was carried out at the SAIC Advanced Research Facility (Frederick, MD, USA). Briefly, human embryonic kidney cells HEK293E (American Type Culture Collection, Manassas, VA, USA) were transfected with 1 mg of plasmid DNA, and supernatants were collected after 72 h of transfection. Recombinant proteins were purified by affinity chromatography followed by size-exclusion chromatography, using Nickel-charged HiTrap Chelating HP and Superdex 200 10/300 GL columns, respectively (GE Healthcare Life Science, Chicago, IL, USA). All protein purification experiments were carried out using the AKTA purifier system (GE Healthcare Life Sciences, Chicago, IL, USA). Purified protein was separated in a NuPAGE Novex 4–12% Bis-Tris protein gels (Thermo Fisher Scientific, Waltham, MA, USA) and visualized by Coomassie stain using the eStain protein stain system (GenScript, Piscataway, NJ, USA). Protein identity was verified by Edman degradation at the Research Technologies Branch, NIAID, NIH.

### 4.2. Surface Plasmon Resonance (SPR) Analysis

All SPR experiments were carried out in a T100 instrument (GE Healthcare Life Science, Chicago, IL, USA) following the manufacturer’s instructions. Sensor CM5, amine coupling reagents, and buffers were purchased from GE Healthcare. HBS-P (10 mM HEPES, pH 7.4, 150 mM NaCl, and 0.005% (*v*/*v*) P20 surfactant) was used as the running buffer for all SPR experiments. All SPR experiments were analyzed using the Biacore Evaluation software v2.0.3 provided by GE Healthcare. Briefly, 15 µg/mL of the envelope protein of ZIKV virus (Cat. Number: DAGA6128, Creative Diagnostics, Shirley, NY, USA) in acetate buffer, pH 4.5, was immobilized over a CM5 sensor via amine coupling. Blank flow cells were used to subtract the buffer effect on the sensogram. Binding experiments were carried out with a contact time of 120 s at a flow rate of 30 µL/min at 25 °C and complex dissociation was monitored for 300 s. Kinetic assay were carried out with a contact time of 180 s at a flow rate of 40 µL/min at 25 °C. and complex dissociation was monitored for 600 s, the sensor surface was regenerated by a 45-s pulse of 50 nM NaOH at 30 µL/min in all SPR experiments.

### 4.3. Binding ELISA Assays

To analyze the binding of mosquito recombinant mosquito salivary proteins with viral proteins, 96 wells flat-bottom plates (Costar, Corning, NY, USA) were coated overnight at 4 °C with 5 μg/mL ZIKV envelope protein or virus-like particles ((ZIKV-VLPs) Cat. Number: ZIKV-VLP-100, The Native Antigen company (Oxfordshire OX5 1LH, United Kingdom), in carbonate bicarbonate buffer pH 9.6 (Sigma, Burlington, MA, USA). Plates were washed with 0.05% *v*/*v* Tween-20 in TBS (25 mM Tris, 150 mM NaCl, pH 7.4) between all steps before using the stop solution. Plates were blocked with blocking buffer (TBS-Tween with 5% BSA) for 1 h at room temperature. After the blocking step, plates coated with ZIKV-VLPs were incubated overnight a 4 °C with the AAEL000793, AAEL007420, or AAEL006347 proteins (10 μg/mL) in PBS. Anti-histidine (His-Tag-Ab (I Invitrogen, Waltham, MA, USA)) at 1:1500 dilution was used as a primary antibody to detect proteins interactions and pan-flavivirus 4G2 antibody (4G2-Ab) at 1:500 dilution was used as a positive control to detect the ZIKV-VLPs. Primary antibodies were incubated for one hour at room temperature. Plates coated with the ZIKV-E protein were blocked under the same conditions described above and then incubated overnight a 4 °C with the AAEL000793 protein (10 μg/mL), mice polyclonal serum AAEL000793-pAbs was used as a primary antibody (1:500 dilution) for 1 h at room temperature. In all the ELISAs binding assays we used, alkaline phosphatase-coupled anti-mouse IgG (1:10,000 in TBS-Tween, (Sigma, Burlington, MA, USA)) was added. Plates were developed with stabilized p-nitrophenyl phosphate (Sigma, Burlington, MA, USA) and absorbance was measured at 405 nm in a VersaMax microplate reader (Molecular Devices) after a 15-min incubation.

### 4.4. Polyclonal Antibodies

Five female BALB/c mice were inoculated with 50 μg of AAEL000793, AAEL007420, AAEL006347 or AAEL003601 recombinant protein in Magic Mouse adjuvant (Creative Diagnostics, Shirley, MA, USA). Mice were boosted three weeks after the primary injection with the same amount of protein. Terminal bleeds were performed at 35 days post first immunization. All injections and terminal bleeds were carried out by the NIAID/NIH animal facility personnel.

### 4.5. Anti- Recombinant Salivary Protein Antibody Detection in Mice Serum Samples Inmunized with the Salicary Recombinants AAEL000793, AAEL007420 and AAEL006347

Antibodies levels were determined by ELISA assays. Briefly, 96-well flat bottom plates (Costar, Corning, NY, USA) were coated with 2 μg/mL of recombinant proteins AAEL000793, AAEL007420, or AAEL006347 diluted in carbonate bicarbonate buffer, pH 9.6 (Sigma-Aldrich, Burlington, MA, USA). Plates were blocked with 5% bovine serum albumin (BSA) in Tris-buffered saline (TBS) (25 mM Tris, 150 mM NaCl, pH 7.4). After three washes with TBS with 0.05% (*v*/*v*) Tween (TTBS), pooled mouse sera immunized with the AAEL000793, AAEL007420 andAAEL006347 recombinant protein were diluted 1:1000 in TTBS then were added to the plates. Following 1 h of incubation and washing, alkaline phosphatase-conjugated anti-mouse IgG (Sigma-Aldrich, Burlington, MA, USA) at dilution 1:10,000 in TTBS was added. After three more washes, plates were developed with stabilized p-nitrophenyl phosphate (Sigma-Aldrich, Burlington, MA, USA) and absorbance was measured at 405 nm in a VersaMax microplate reader (Molecular Devices, San Jose, CA, USA).

### 4.6. Anti- Recombinant Salivary Protein Antibody Detection in DENV and ZIKV Serum Samples

Recombinant mosquito salivary proteins AAEL000793, AAEL007420, or AAEL006347 (2 μg/mL) were coated on a 96-well, flat bottom plate (Costar, Corning, NY, USA) overnight at 4 °C. Wells were washed three times in TBS-Tween a blocked with blocking buffer (2% BSA in TBS-T). After 1 h, wells were washed three times in TBS-Tween incubated with ZIKV or DENV serum samples (1:200 dilution) in blocking buffer for 1 h. Wells were washed three times and incubated with HRP anti-human and IgG H+L (1:10,000 dilution in blocking buffer). Colorimetric analysis was performed by measuring absorbance values at 450 nm in a VersaMax microplate reader (Molecular Devices, San Jose, CA, USA).

### 4.7. Cells and Viruses

Primary Epidermal Keratinocytes from Neonatal Foreskin (HEKn) and Primary Dermal Microvascular Endothelial Cells (HDMVECn) were purchased from ATCC^®^ (Gaithersburg, MD, USA (Cat. Number: PCS-200-010 and PCS-110-010)) and maintained for up to six passages. Cells were grown in 37 °C following ATCC’s methods. The ZIKV virus (ZIKV) was propagated in C3/36 cells derived from *Aedes albopictus*. These cells were grown in minimum essential medium (pH 7.2) supplemented with 10% fetal bovine serum (FBS), sodium pyruvate, l-glutamine, and nonessential amino acids (NEAA) (all from Gibco) and incubated at 37 °C. ZIKV stocks were prepared by infecting a monolayer of C6/36 cells with 80–85% confluence for 24 to 48 h. The supernatant was collected, cells were cleared by centrifugation at 1000× *g* for 10 min at 4 °C and concentrated in 0.22-μm Amicon columns to 1/10 of the starting volume. The stocks were aliquoted and stored at −80 °C until further use.

### 4.8. ZIKV Virus Quantification

Virus titers of the supernatant were determined by plaque assay. Vero cells ATCC^®^ (Gaithersburg, MD, USA (Cat. Number: CCL-81, ATCC^®^)) were seeded at a density of 1 × 10^5^ cells per well in standard 24 well plates. Serial dilutions of culture supernatant were added and incubated for 2 h at 37 °C. The cell monolayers were then overlaid with 1 mL/well of Opti-MEM (Gibco, Grand Island, NY, USA) containing 1% methylcellulose, 2% fetal bovine serum (FBS), 2 mM L-glutamine, and 1X antibiotic-antimycotic solution (Gibco, Grand Island, NY, USA). Plates were incubated at 37 °C and 5% CO_2_ for 4–5 days. Cells were washed three times with PBS and stained with 1 mL of naphthol blue-black solution (1 g of naphthol blue black, 13.6 g of sodium acetate, 60 mL glacial acetic acid in 1 L of distilled and deionized H_2_O) for 30 min for plaque visualization.

### 4.9. Serum Samples from DENV and ZIKV Patients

Convalescent serum samples from ZIKV-infected patients were obtained from BEI Resources, NIAID, NIH. Serum samples from DENV-infected patients were obtained from Dr. Berlin Londono (Department of Entomology, Kansas State University, Manhattan, KS, USA), from a separate cohort study conducted in at the Los Patios Hospital, Cucuta, Norte de Santander, Colombia. Serum collections and diagnostics performed in these samples have been described previously by her group [35].

### 4.10. Mosquito Salivary Gland Dissections

*Aedes aegypti* (Liverpool strain) mosquitoes were reared under standard conditions in a 12-h light/dark cycle at and maintained at 28 °C, 80% humidity, at the Laboratory of Malaria and Vector Research insectary, NIAID, NIH. Salivary glands from sugar-fed 5- to 8-day-old adult female mosquitoes were dissected in ice-cold PBS pH 7.4 using a stereomicroscope (Zeiss, Thornwood, NY, USA). Next, salivary gland extracts (SGE) was obtained by sonicating dissected salivary glands in PBS, pH 7.4 using a Branson Sonifier 450 (Branson Ultrasonics, Danbury, CT, USA). Supernatants were recovered after disrupted tissues were centrifuged for 5 min at 12,000× *g*. The concentration of protein content was measured by spectrophotometry at A_280_ (DS-11, DeNovix, Wilmington, DE, USA), then the SGE was kept at −80 °C until use.

### 4.11. Statistical Analysis

Data were analyzed by using GraphPad Prism v 7 software and plotted as bar graphs or scatter plots. Results were analyzed by Mann-Whitney test, one-way or two-way ANOVA, statistical differences were set at *p* < 0.05.

## Figures and Tables

**Figure 1 viruses-14-00221-f001:**
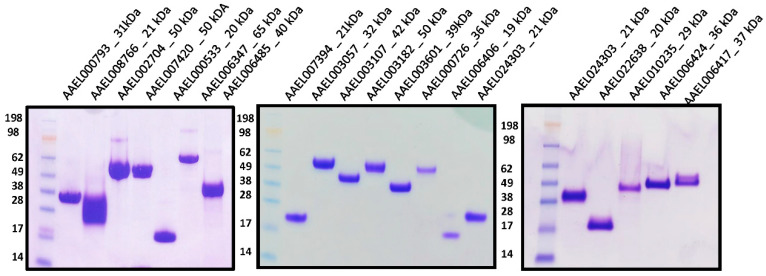
Purification of *Aedes aegypti* salivary recombinant proteins. All the recombinant proteins purification were confirmed by Coomassie-stained NuPAGE 4–12%. A single band was visualized in each protein. See Blue Plus2 Pre-stained protein ladder was used as protein standards. Protein identity was confirmed by N-terminal sequencing.

**Figure 2 viruses-14-00221-f002:**
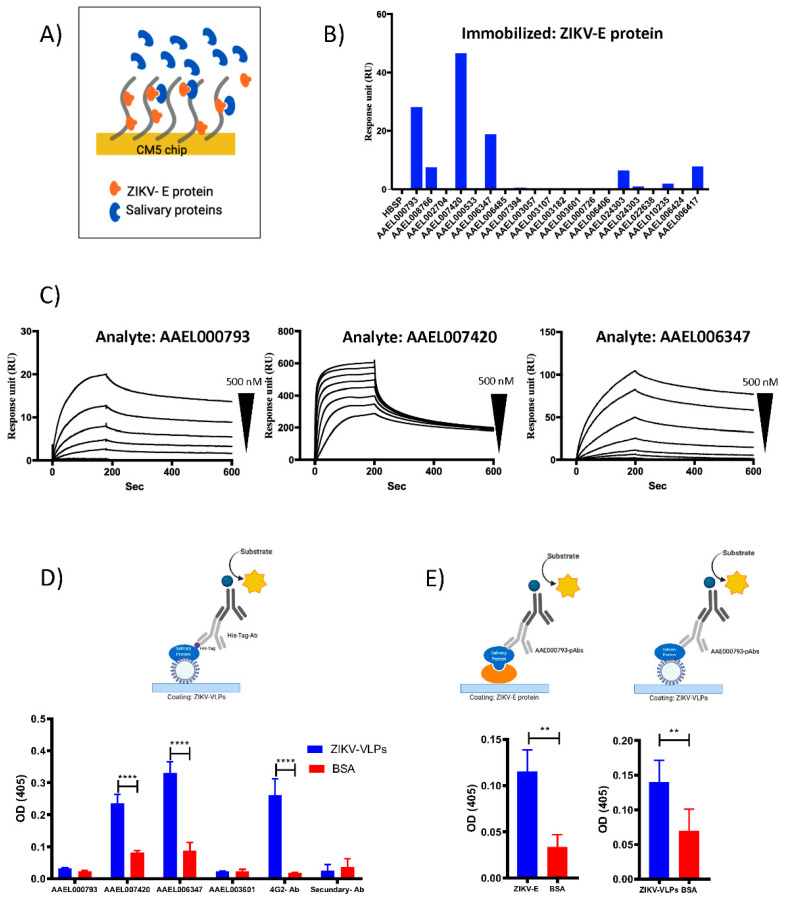
*Aedes aegypti* proteins bind to ZIKV-E and ZIKV-VLPs. (**A**) Schematic experimental set-up of surface plasmon resonance assay. (**B**) Surface plasmon resonance binding analysis, *Aedes aegypti* recombinant salivary proteins (concentration 200 nM), were flowed over 15 μg/mL of immobilized ZIKV-E protein. (**C**) Kinetic of the interaction of recombinant AAEL000793, AAEL007420, and AAEL006347 over ZIKV-E protein. Recombinant salivary proteins were flowed over immobilized ZIKV-E protein for 180 s at 40 μL/mL, and the dissociation of the protein was monitored for 600 s. The response data were fitted to a 1:1 interaction model using global analysis. (**D**) Schematic representation of ELISA binding assay and results. ZIKV-VLPs (5 μg/mL) were immobilized and incubated with the recombinant AAEL000793, AAEL007420, AAEL006347 (10 μg/mL), the interaction between ZIKV-VLPs and recombinant salivary proteins was detected using His-Tag-Ab. Pan-flavivirus 4G2 antibody (4G2-Ab) was used as a positive control to detect the immobilized ZIKV-VLPs. (**E**) Schematic representation of ELISA binding assay and results, ZIKV-E protein or ZIKV-VLPs at 5 μg/mL were immobilized and incubated with the recombinant AAEL000793 (10 μg/mL) polyclonal antibodies AAEL000793-pAbs was used to detect the interaction between proteins. Results are indicated as the mean ± SD. Asterisks show statistical significance compared with the BSA control (*p* values are indicated: **: *p* < 0.01; ****: *p* < 0.0001.). Schematic representations were generated with BioRender.com (created on 12 May 2021).

**Figure 3 viruses-14-00221-f003:**
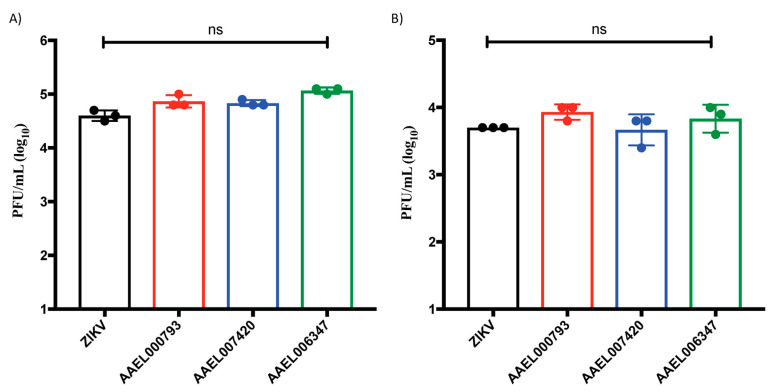
Salivary recombinant proteins AAEL000793, AAEL007420, and AAEL006347 do not affect the viral replication in cultured endothelial cells and keratinocytes. (**A**) HDMVECn and (**B**) HEKn cells were infected with ZIKV virus at 5 MOIs during 24 h. Infected cell supernatants were analyzed by plaque assay to quantify infectious viral particles. Statistical analyses were done from three independent experiments each performed in triplicate (ns: no significant).

**Figure 4 viruses-14-00221-f004:**
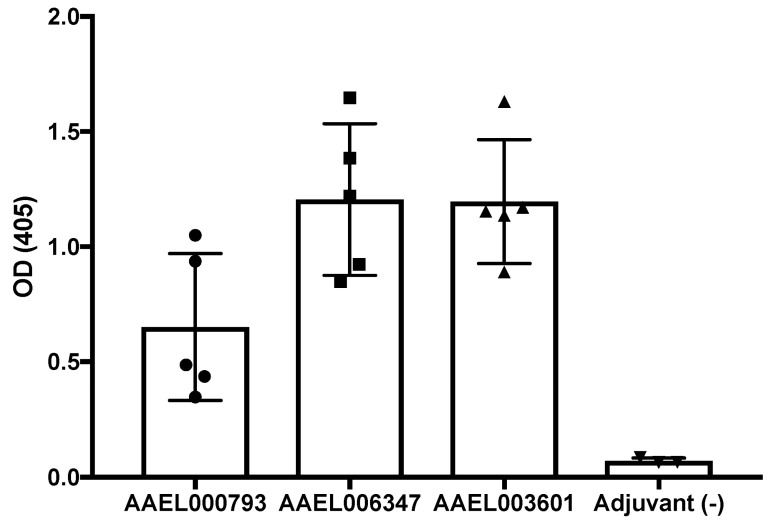
Recombinant salivary proteins AAEL000793, AAEL006347 and AAEL003601 induce antibody response in BALB/c mice. Six-week-old female BALB/c mice (5 per group) were inoculated with 50 µg of recombinant AAEL000793, AAEL006347 or AAEL003601. An enzyme-linked immunosorbent assay (ELISA) was conducted following the final bleed to measure the resulting IgG antibody levels, and any cross-reactivity between proteins and their specific antibodies. Antibodies levels were determined by ELISA assays. Mice immunized with Magic Mouse Adjuvant alone were used as a negative control. Bars show means of optical density values of all the proteins. OD = optical density.

**Figure 5 viruses-14-00221-f005:**
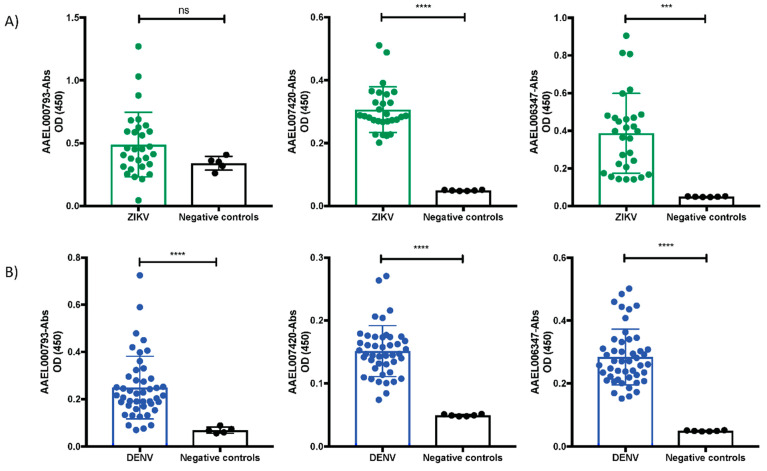
Levels of IgG serum antibodies against the salivary proteins in patients with DENV and ZIKV disease. ELISA assays were performed using serum samples at dilution 1:200 from ZIKV (**A**) and DENV patients (**B**). Serum dilutions were incubated over ELISA plates coated with 2 μg/mL of AAEL000793, AAEL007420, or AAEL006347 salivary proteins. Results are indicated as the mean ± SD. Asterisks show statistical significance comparing with the control serum samples (*p* values are indicated: ns: no significant; ***: *p* < 0.001; ****: *p* < 0.0001).

**Table 1 viruses-14-00221-t001:** Surface plasmon resonance analysis of salivary proteins and ZIKV-E protein interaction.

Protein	ka (1/Ms)	kd (1/s)	KD (nM)
NIH435-3	20,400 ± 4667	0.0005310 ± 0.00004243	26.49 ± 4.101
NIH435-7	5,681,850 ± 7,945,264	0.003508 ± 0.002630	13.18 ± 17.97
NIH435-9	75,550 ± 1485	0.001057 ± 0.0003019	13.99 ± 4.243

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
