# Peer review of "Multiple Salivary Proteins from Aedes aegypti Mosquito Bind to the Zika Virus Envelope Protein"

_viruses, 2022, doi:10.3390/v14020221_

Round 1

Reviewer 1 Report

The authors have identified 3 protéines of Aedes aegypti saliva able to bind the envelope protein of Zika virus. Their work consisted first in expressing 21 saliva proteins, which were screened by SPR for binding to the virus envelope. The binding proteins were further tested by ELISA. These proteins were found to be immune reactive by testing the sera of patients infected by ZIKV and DENV. Finally, they tested whether antibodies that recognized the most immunogenic salivary protein affected the ADE.

The summary is concise and accurately reports the main results obtained. however, the effect of anti-salivary protein antibodies on ADE is expected to be more moderate given the small increase in infected cells observed. The introduction is clear and well documented. It would have been interesting to also cite the work of Cao-Lormeaux 2009 which highlights several salivary proteins capable of interacting with the dengue virus. Regarding the production of salivary proteins, the authors must justify their choice of eukaryotic cells rather than arthropod cells. the glycosylations are not the same and could thus impact the interaction with the virus. The screening which was carried out using Biacore is interesting but the results deserve to be clarified. In FIG. 2B, the 3 proteins of interest are all used at a concentration of 200 nM and give relatively homogeneous responses of between 20 and 45 RU. On the other hand, in FIG. 2C, when cascading dilutions are carried out between 500 nM and 0.015nM, large differences in RU are noted between the proteins. How can we justify these differences? Regarding ELISA, I think that a competition approach would have been interesting to achieve to ensure the specificity of the interaction. This would consist in varying the concentration of the protein of interest on the coating in order to determine a concentration equivalent to an interaction equal to 50% of the value of the plateau. This concentration would then have been used in preincubation with variable concentrations of VLP then the mixture would have been incubated on the plate. this allows to have a real answer on the specific interaction.

Paragraph 2.4 I do not understand the title. This is not to say that antibodies are immunogenic, but proteins are. Could you modify the title accordingly?

Paragraph 2.5 I would like Supplementary Figure 2 to be shown in the text and Figure 5 which is not very convincing to be an additional figure. Are K562 cells permissive to Zika virus. This is not specified in the text and in the Math Meth but it is important to understand the results obtained. Why did the authors use salivary gland extracts and not the proteins of interest? The effect observed in the presence of salivary gland extract in FIG. 5A is very weak and is not physiologically significant. As for the "trend" that the authors refer to in Figure 5B, I do not see it well and given the small number of samples tested, it should not appear in the results. Moreover the human sera that were used in these experiments need to be described in more detailed. We do not know what means that they induce ADE at high or low concentration. Could you please add these details in the Math Meth

Similarly in the discussion, I don't think the results on ADE are convincing enough and need to be much more moderate than in lines 236. 237

lines 268-269 the explanation is not clear. Do the authors mean that the tag is masked during the interaction with ZIKV thereby hindering the binding of anti-Tag antibodies

In general the idea that salivary proteins may influence the ADE is attractive but not sufficiently supported by the results obtained. The experiments must be redone with more human serum samples.

Author Response

Comments and Suggestions for Authors

We thank the reviewer for identifying these errors and for her/his helpful recommendations. The reviewer’s comments are plain font, and our responses are in bold font

The authors have identified 3 protéines of Aedes aegypti saliva able to bind the envelope protein of Zika virus. Their work consisted first in expressing 21 saliva proteins, which were screened by SPR for binding to the virus envelope. The binding proteins were further tested by ELISA. These proteins were found to be immune reactive by testing the sera of patients infected by ZIKV and DENV. Finally, they tested whether antibodies that recognized the most immunogenic salivary protein affected the ADE. 

The summary is concise and accurately reports the main results obtained. however, the effect of anti-salivary protein antibodies on ADE is expected to be more moderate given the small increase in infected cells observed.

The introduction is clear and well documented. It would have been interesting to also cite the work of Cao-Lormeaux 2009 which highlights several salivary proteins capable of interacting with the dengue virus.

We have added this reference to the manuscript.

Regarding the production of salivary proteins, the authors must justify their choice of eukaryotic cells rather than arthropod cells. the glycosylations are not the same and could thus impact the interaction with the virus.

We understand the concern raised by the referee, but we believe the system used for this project has not significantly impacted the results presented here. Heterologous expression of blood feeding arthropod (mosquitoes, ticks, sandflies, etc) recombinant proteins in eukaryotic (HEK cells) and prokaryotic (E. coli) systems is well documented and accepted in the literature (reviewed in Curr Opin Insect Sci. 2018 Oct;29:102-109; Parasite Immunol. 2021 May;43(5):e12807, and others). Transient expression in mammalian cells is very quick and can give expression as high or even higher than with insect cells especially for secreted proteins. The fact that only 3 proteins bound to E-protein and VLP, suggests that this is not a non-specific binding activity. Also, there are several examples in the literature using HEK293 expression system where authors tested novel salivary activities with deglycosylated version of recombinant proteins to demonstrate that mammalian glycosylation did not have any negative effect on the studied activities (Arterioscler Thromb Vasc Biol. 2012 Sep;32(9):2185-98; PLoS Pathog. 2014 Feb 6;10(2):e1003923; Nat Commun. 2021 May 28;12(1):3213).

The screening which was carried out using Biacore is interesting but the results deserve to be clarified. In FIG. 2B, the 3 proteins of interest are all used at a concentration of 200 nM and give relatively homogeneous responses of between 20 and 45 RU. On the other hand, in FIG. 2C, when cascading dilutions are carried out between 500 nM and 0.015nM, large differences in RU are noted between the proteins. How can we justify these differences?

For these 2 experiments, based on their experimental design (sensor surface density, flow rate, and contact time) are different, and one would expect to see variability in RU between binding assay and kinetics. These differences in experimental design justify the RU levels in Figures 2B and C. The results shown in Figures 2B and C and are typical of SPR experiments. We have expanded the description of running conditions for kinetic experiments in Materials and Methods section for clarification.

Regarding ELISA, I think that a competition approach would have been interesting to achieve to ensure the specificity of the interaction. This would consist in varying the concentration of the protein of interest on the coating in order to determine a concentration equivalent to an interaction equal to 50% of the value of the plateau. This concentration would then have been used in preincubation with variable concentrations of VLP then the mixture would have been incubated on the plate. this allows to have a real answer on the specific interaction.

Although we agree with the referee, the ELISA with VLP was used as an orthogonal method to validate the binding of the mosquito recombinant salivary proteins to the E-protein can also occur in VLP. We did not intend to calculate any kinetic values or comparing binding activities among the 3 recombinant proteins to VLP. A recombinant protein with no binding to E-protein in SPR was used as a negative control in VLP ELISA. Our findings agree with a newly published article by Gavor et al (Life Sci Alliance. 2022 Jan; 5(1): e202101211). This work is relevant to our finding since it describes another mosquito protein biding directly to DENV-2 VLPs by ELISAs assays. This reference is now cited in our work.

Paragraph 2.4 I do not understand the title. This is not to say that antibodies are immunogenic, but proteins are. Could you modify the title accordingly?

We apologize for the typo. We have changed it to: Mosquito salivary proteins interacting with ZIKV-E and ZIKV-VLPs virus are highly immunogenic in patients with ZIKV and DV. This is also an indirect indication that these proteins are secreted and injected in human hosts during blood feeding.

Paragraph 2.5 I would like Supplementary Figure 2 to be shown in the text and Figure 5 which is not very convincing to be an additional figure

Thank you for this suggestion. We have moved supplementary figure 2 in the main figures.

Are K562 cells permissive to Zika virus. This is not specified in the text and in the Math Meth but it is important to understand the results obtained. Why did the authors use salivary gland extracts and not the proteins of interest?

K562 cells are poorly permissive for infection in the absence of antibodies and K562 is a widely used ADE activity of anti-flavivirus antibodies. K562 express FcγR and can internalize the viruses in complex with anti-virus antibodies (Littua et al. J Immunol. 1990 Apr 15;144(8):3183-6). We have added this description and reference in the Materials and Methods section.

We used SGE extract in Figure 5 (now moved to Supplementary Figure) for as a detection reagent for this study because the scarce number of human samples available to us. Also, SGE could be closer to what will happen during natural viral transmission in the field. It has also been demonstrated that Ae. aegpyti SGE exacerbates systemic dengue pathogenesis during ADE (Schmid et al. PLoS Pathog. 2016 Jun 16;12(6):e1005676). Please, also see the last paragraph in this letter in response to your comments.

The effect observed in the presence of salivary gland extract in FIG. 5A is very weak and is not physiologically significant. As for the "trend" that the authors refer to in Figure 5B, I do not see it well and given the small number of samples tested, it should not appear in the results. Moreover, the human sera that were used in these experiments need to be described in more detailed. We do not know what means that they induce ADE at high or low concentration. Could you please add these details in the Math MethSimilarly in the discussion, I don't think the results on ADE are convincing enough and need to be much more moderate than in lines 236. 237

We agree and thank the referee for his/her suggestion. To better reflect the main theme of our work, we have changed the title to “Multiple salivary proteins from Aedes aegypti mosquito bind to the Zika virus envelope protein. We have moved Figure 5 to supplementary materials. Materials and Methods section was edited to accommodate the referee’s suggestion (extensively detailed in PLoS One. 2013 Dec 2;8(12):e81211). We have also soften/moderate the discussion in lines 236-237.

lines 268-269 the explanation is not clear. Do the authors mean that the tag is masked during the interaction with ZIKV thereby hindering the binding of anti-Tag antibodies.

This correct. It appears that the 6x-His tag is masked under native conditions (but accessible under denatured conditions). For native condition experiments, we used polyclonal antibodies raised against the recombinant protein. We have modified lines 268 and 269 to better explain this observation.

In general, the idea that salivary proteins may influence the ADE is attractive but not sufficiently supported by the results obtained. The experiments must be redone with more human serum samples.

We concur with the referee that more samples are needed to confirm our findings. However, we do not have a better-cataloged sample pool to carry out a more in-depth study at this time. The results presented here are preliminary and exploratory to pave the way to a large broader study in the field. We are currently collecting field samples in Cambodia for this work. We have made several changes in the manuscript to “moderate” the reach of out conclusions. We have also changed the title to “Multiple salivary proteins from Aedes aegypti mosquito bind to the Zika virus envelope protein”.

Reviewer 2 Report

Summary:

The manuscript by Paola et al. has evaluated the interactions of Aedes salivary proteins and ZIKV and identified three proteins of AAEL000739, AAEL007420 and AAEL006347 with relatively higher binding affinities with envelope protein of ZIKV by SPR and ELISA assays. The ADE of antibodies against the salivary protein and virus complex were further evaluated in vitro. Overall, the paper is well-written and easy to follow, and the experiments is accurately designed with both positive and negative controls, while I do have quite a number of comments/suggestions for further improvements.

  • Line 82 “the selection criteria included presence of secretion sequence and expression in the salivary gland of female mosquito only”. Is that possible some other salivary proteins expressed in both sexes might have functions while being overlooked with this selection criteria? Also, the RT-PCR data for selection is strongly recommended in supplementary materials.
  • In 2.3 Section (Line), the data showed no viral replications were observed in two types of cell cultures when proteins interacting with ZIKV. I found that only one condition (24hrs infection along with 200nM protein concentrations) was actually conducted in experiments. It is kind of weak to evidence the conclusion. To address this question, my suggestions is to provide more solid data (different infection duration, different of protein concentrations, and etc.) to make the conclusion strong. Also, the immunofluorescence staining assay and microscopy would be helpful to better illustrate the binding and entry of complex into cells.
  • In Section 2.5, the ADE was tested in vitro. According to the title, ADE should be the main theme of this manuscript, while one figure solely looks not strong enough. This section looks inconsistency in terms of the overall storytelling structures. The ADE evaluated in two antibody concentrations revealed two different scenarios. It has been known the lower concentrations is  more easily to trigger ADE. My question is how you decided to use these two concentrations to test? Have a range of different conc. been tested in your preliminary experiments? Besides, the difference is not significant at higher conc. based on the statistic analysis.

Author Response

Comments and Suggestions for Authors

We thank the reviewer for identifying these errors and for her/his helpful recommendations. The reviewer’s comments are plain font, and our responses are in bold font.

The manuscript by Paola et al. has evaluated the interactions of Aedes salivary proteins and ZIKV and identified three proteins of AAEL000739, AAEL007420 and AAEL006347 with relatively higher binding affinities with envelope protein of ZIKV by SPR and ELISA assays. The ADE of antibodies against the salivary protein and virus complex were further evaluated in vitro. Overall, the paper is well-written and easy to follow, and the experiments is accurately designed with both positive and negative controls, while I do have quite a number of comments/suggestions for further improvements.

Line 82 “the selection criteria included presence of secretion sequence and expression in the salivary gland of female mosquito only”. Is that possible some other salivary proteins expressed in both sexes might have functions while being overlooked with this selection criteria? Also, the RT-PCR data for selection is strongly recommended in supplementary materials.

We agree it’s possible that in our selection criteria, some proteins have been overlooked; we believe that salivary protein present in male and female mosquitoes are injected into the host during blood feeding. A paper published by Peng et al. (Int Arch Allergy Immunol. 2016; 170 (3): 206-10) found that Aed A4 is a major allergen in mosquito saliva, and the protein is also present in male and female mosquitoes. Based on these findings, it would be interesting to investigate the role in pathogen transmission of other salivary proteins present in both sexes. However, we focused on female-specific proteins. Like many blood-feeding arthropods, male mosquitoes do not feed on blood and may not be directly involved in pathogen transmission. This sex and tissue specificity allowed us to narrow down candidate proteins with possible function (s) in virus transmission. We have added the following paragraph in Results: Like many blood-feeding arthropods, male mosquitoes do not feed on blood and may not be directly involved in pathogen transmission. We focused our efforts on salivary proteins expressed in female mosquitoes. This sex and tissue specificity allows us to narrow down candidate proteins with possible pathogen transmission functions.

We did not carry out the RT-PCR experiments. Our selection of female specific genes was based on the published data by Ribeiro et al (PLoS One, 2016;11(3):e0151400).

In 2.3 Section (Line), the data showed no viral replications were observed in two types of cell cultures when proteins interacting with ZIKV. I found that only one condition (24hrs infection along with 200nM protein concentrations) was actually conducted in experiments. It is kind of weak to evidence the conclusion. To address this question, my suggestions is to provide more solid data (different infection duration, different of protein concentrations, and etc.) to make the conclusion strong.

Indeed, we did not find any significantly difference in viral replication using keratinocytes or endothelial cells. We also tried 100nM and 200nM at 24hrs and 48 hrs post-infection (not shown) in endothelial cells and neither concentration nor time showed any difference in viral replication. We selected 24 hrs. post-infection based on previously published work using keratinocytes and WNV and DV (Pornapat Surasombatpattana et al. J Invest Dermatol. 2014 Jan;134(1):281-284; Magali Garcia et al. Front Cell Infect Microbiol. 2018 Nov 2;8:387). These authors show that Aedes aegypti saliva or Aedes recombinant salivary proteins affect viral replication at 24 hrs by RT-qPCR.

Also, the immunofluorescence staining assay and microscopy would be helpful to better illustrate the binding and entry of complex into cells.

Because we did not detect any viral replication in the presence of the recombinant proteins, we did not carry out additional experiments to visualize cell entry of the virus. In the case of ADE, we utilized flow cytometry to quantitate viral entry into K562 cells.

In Section 2.5, the ADE was tested in vitro. According to the title, ADE should be the main theme of this manuscript, while one figure solely looks not strong enough. This section looks inconsistency in terms of the overall storytelling structures. The ADE evaluated in two antibody concentrations revealed two different scenarios. It has been known the lower concentrations is more easily to trigger ADE. My question is how you decided to use these two concentrations to test? Have a range of different conc. been tested in your preliminary experiments? Besides, the difference is not significant at higher conc. based on the statistical analysis.

We agree and thank the referee for his/her suggestion. To better reflect the main theme of our work, we have changed the title to “Multiple salivary proteins from Aedes aegypti mosquito bind to the Zika virus envelope protein”.

We apologize for the confusion in the ADE and antibody concentrations used in Figure 5. All (8) samples were tested in 4 different concentrations. Figures 5 A and B are consolidated in now Supplemental Figure 2 as recommended by the referee.

We also agree with the referee’s point regarding lower concentration of neutralizing antibodies may trigger ADE. However, DV serum samples have poorly neutralizing activity in ZIKV infections at high concentrations (Wanwisa Dejnirattisai et al. Nat Immunol 2016 Sep;17(9):1102-8). We also used DV serum samples for our work and this might explain our findings.

We also concur with both referees’ opinion that more samples are needed to confirm our findings regarding ADE and anti-mosquito salivary proteins antibodies. However, at this time, we do not have a better-cataloged sample pool to carry out a more in-depth study. The results presented here are preliminary and exploratory to pave the way to a large, broader study in the field. We are currently collecting field samples in Cambodia for a follow-up work. Based on the reviewers’ suggestions, we have moved ADE experiments to supplementary data since this is not the main theme of our work.

Round 2

Reviewer 1 Report

The authors did take into account all the comments and modified their manuscript. The new version is as I expected and can be published

Author Response

Thank you!

Reviewer 2 Report

I think the authors have improved the manuscript and clarified the main theme by modifying the title. I do not have other comments on it.

Author Response

Thank you!